# A Spiritual Theology of Integral Human Development: To "Grow in Holiness"

Glenn Joshua Morrison 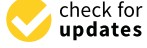

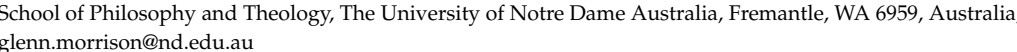

School of Philosophy and Theology, The University of Notre Dame Australia, Fremantle, WA 6959, Australia; glenn.morrison@nd.edu.au

**Abstract:** The article identifies the nature of integral human development as a Christian imperative and an incarnational life of responsibility for others. To grow in holiness through the truth of the Gospel signifies overcoming the egoism of the self, being generous in responsibility (love in truth), and discovering a beatitude of hope to become sons and daughters of God (truth in love). Engaging truth in the light of history, evil, and death, the article proceeds to relate the encounter of the soul with "the depths of God" (1 Cor 2:10) to learn from the Spirit a life aimed for the common good. The path to "the depths of God" is one of hope to encounter the vulnerability of the other and oneself, a journey into boldness, newness, and redemption with Christ towards the face of the forsaken and poor. Integral human development, a pathway of peace and healing "to the far and the near" (Isa 57:19), is otherwise than an evasion of love and responsibility. For in the proclamation and witness that "God is love" (1 Jn 4:16) lies the hope to build the earthly city of God and herald an end to war, indifference, and hatred of others.

**Keywords:** integral human development; eschatology; Levinas; love; Maritain; otherness; Pope Benedict XVI; spirituality; truth; von Balthasar

## 1. Introduction

Integral human development is borne out of the Catholic Intellectual Tradition animated by the nineteenth-century growing concerns for social justice and the twentieth-century developments of religious-inspired humanism. In Jacques Maritain's words, "the *concrete logic of the events of history*" (Maritain 1936, p. 1) invited a fraternal, social response. After Vatican II, responding to world interest in "development" (Pope 2019, p. 125), key papal writings began to emerge and engage "development" to give a theological and spiritual witness to social and political concerns so that "the rational creature should of his own accord direct his life to God, the first truth and the highest good" (Pope Paul VI 1967, no. 16). Later, Pope St. John Paul II would assert, "If 'development is the new name for peace,' war and military preparations are the major enemy of the integral development of peoples" (Pope John-Paul II 1987, no. 10). Less than ten years later, Pope Benedict XVI enumerated, "Love in truth—*caritas in veritate*—is a great challenge for the Church in a world that is becoming progressively and pervasively globalized" (Pope Benedict XVI 2009, no. 9). And more recently, commenting from the context of a growing alienation through a "consumer society" of "feverish demands . . . superficial information, instant communication, and virtual reality" (Pope Francis 2018, no. 108), Pope Francis stated, "In such a society, politics, mass communications and economic, cultural, and even religious institutions become so entangled as to become an obstacle to authentic human and social development. As a result, the Beatitudes are not easy to live out; any attempt to do so will be viewed negatively, regarded with suspicion, and met with ridicule" (Pope Francis 2018, no. 91).

Together, these papal voices testify to the essential hope for "integral human development" in a globalized world rapidly embracing "superficial" and morally evil ways of living and relating (Pope Francis 2018, no. 108). Such a hope resounds in the language of

fraternity, solidarity, generosity, and spiritual formation and growth. Aspiring towards such a language of transcendence amongst people of faith is a first step to meeting and responding to the great dangers to human flourishing and integral development, such as war and genocide, alienation, loneliness, superficiality, indifference to suffering, hunger, and poverty. A major challenge then is to hear God's word as much as act upon it.

In terms of introducing and developing a spiritual theology of integral human development, the Christian imperative, "Truly, I tell you, just as you did it to one of the least of these who are members of my family, you did it to me" (Matt 25:40), offers a directive to form humanity in the goodness of truth and love, to hear and respond to God's word in the other's face. Further, for example, having walked centuries to splinter in hope and collide with human reason, the Gospel imperative, "You shall love your neighbor as yourself" (Matt 22:39), has set out to put into question history and culture as much as economic, social, and political consciousness. The "concrete logic of the events of history" in all its "human wisdom" (1 Cor 1:25) remains a challenge to integral human development. This is because, by inviting a spiritual theological and biblical lens, integral human development attunes to the "stumbling block" and foolishness of "Christ crucified" (1 Cor 1:23).

Integral human development takes place in and with Christ (2 Cor 6:4–5), witnessing to a spiritual practice of resilience in faith: "as sorrowful, yet always rejoicing; as poor, yet making many rich; as having nothing and yet possessing everything" (2 Cor 6:10). "God's foolishness" and "God's weakness" (1 Cor 1:25) signifies that "now," today, is the time to proclaim "a day of salvation" (2 Cor 6:2), a day of divine goodness and generosity, through embracing the incarnational and paschal truth that "God is love" (1 Jn 4:16). The little goodness and sensibility of integral human development, borne out of charity in truth, testifies to the invincibility of God's presence in spiritual and material poverty, righteousness, gentleness, mercy, peace, and boldness (cf. Matt 5:3–12). One may call this hope an eschatological witness of faith in God's word as one encounters the poverty and suffering of others.

Integral human development breaks open the integrity, dignity, and goodness of the human person. God enters the inner heart and soul of life, breathing the Spirit of hope to unveil the gift of forgiveness in Jesus Christ. A spirituality of integral human development will invite participation in the mysteries of the faith, to be "poor in spirit" (Matt 5:3), gentle yet bold to accept a pathway towards mission, charity, and truth. To this end, the article itself will seek to develop a spiritual theology of integral human development through a dialogue and conversation between Christian theology and Emmanuel Levinas' philosophy. Levinas, a Jewish–French Talmudic philosopher, offers a radical humanism of the other (alterity) that can help direct integral human development into an ethical, metaphysical domain. At the same time, it invites a provocative challenge to the Christian spiritual and theological imagination of faith to seek not just "a better world," but a "new world" (Kasper [1977] 1993, p. 96), a "wisdom of love at the service of love," or in more theological terms, a "Kingdom of a non-thematizable God" (Levinas [1998] 1999, pp. 52, 162).

By "non-thematizable," Levinas means, "The good that reigns in its goodness cannot enter into the present of consciousness," that is to say, God's goodness ("Kingdom") is beyond the ontological strivings of the "history of Being" (Levinas [1998] 1999, p. 52). The good is beyond being; otherwise, it is anarchic (without origin), immemorial (since or "in" the "beginning" (Gen 1:1)), and diachronic (coming to time through responsibility). Moreover, such goodness, bearing witness to the glory of the Infinite God (that "the Kingdom of God is ethical" (Levinas [1998] 1999, p. 183)), cannot be reduced to the totality of ego-consciousness. Levinas signifies such testimony of divine goodness as the trace of illeity ("the *he* in the depth of the you" (Levinas 1998a, p. 165) that "overflows both cognition and the enigma through which the Infinite [God] leaves a trace in cognition" (Levinas [1998] 1999, p. 162). Illeity is also "holiness" or the spontaneity of responsibility in the words, "Here I am" (Levinas [1998] 1999, pp. 152, 162), namely of having encountered God's word and command to be responsible for the other from the depths of the soul. Illeity can further be conceived as moments of justice overflowing with mercy remaining as a

trace, "a past that has never been present" (Levinas 1998a, p. 165). The idea of charity, in truth, is a "difficult freedom" (Levinas 1990, p. 272) of being held "hostage" (Levinas 1996a, p. 91) to the other and the Infinite. For Levinas, the illeity of God's trace and word reveals a command, ordination/order (Levinas 1998b, p. 111), and "liturgy" (Levinas 1990, p. xiv) of responsibility.

Pursuing Levinas' ethical metaphysics of otherness, the article will engage the following areas to introduce a spiritual theology of integral human development: (i) History and truth; (ii) truth beyond evil and death; (iii) into the "depths of God"; (iv) towards a theology of hope; and (v) to be sons and daughters of God. History need not develop into the convenience of lies. Truth acts to interrupt and contradict the totality of egoism in the face of charity, justice, and mercy. Here, where truth and love come together, a theology of hope comes to mind, that is to say, an eschatological vocation and mission to be sons and daughters of God, to be the leaven (Matt 13:33) in the world, unveiling fragments (Matt 15:27), spaces for healing and compassion, from the depths of the Father's Kingdom to build the foundations for an "*earthly city*" of "unity and peace" (Pope Benedict XVI 2009, no. 7).

For the most part, integral human development has had a greater focus on "Catholic social ethics" (Pope 2019, p. 123). In this article, the aim is to recover and develop the emphasis on the spiritual dimension of integral human development. Given that "Jacques Maritain's 'integral humanism' provided the normative perspective from which the pope [Pope Paul VI] could elaborate a rich and differentiated conception of authentic development" (Pope 2019, p. 127), there are pressing areas of Maritain's emphasis on spiritual, theological themes to begin to put into service. In addition to this dimension, Levinas' ethical metaphysics will also be employed. His writings are not easy to understand. This is because he taxes, as it were, the philosophical and theological imagination of faith to grow toward an ethical horizon of otherness, being for the other. This is at once both demanding and provocative. Levinas demands attention to the sufferings of the other to provoke a response of compassion, the bodiliness of suffering for the other revealing truth as an encounter with God's word. Truth, abiding in God's word, appeals through the other's face and awakens the conscience to act beyond the self-interest of being. Otherwise than the essence of being (as anonymous existence), Levinas situates truth in love beyond eros (the totality of the ego seeking its own possibilities and enjoyment) and articulates love in truth as the hyperbolic giving of oneself for the other. This means, in effect, that responsibility for the neighbor never ends, paralleling that human beings are made in the image of "The Infinite," God.

Levinas' writings have resonated through the work of a growing number of Christian theologians, beginning with such prominent figures as "Jean-Luc Marion, Michel Henry, and Jean-Louis Chrétien in France, and Roger Burggraeve in Belgium" and "of the new generation of theologians and philosophers of religion who advance theology in a phenomenological voice" (Purcell 2006, p. 3). This article will engage especially Levinas' writings with Maritain, von Balthasar, and Popes Benedict XIV and Francis. Von Balthasar will find appeal in the section on eschatology (theology of hope), while Maritain, Pope Benedict XVI, and Pope Francis, writers for the most part who have contributed towards a spiritual and theological understanding of integral human development (Pope 2019, p. 139), will be major dialogue partners. The spiritual and theological papal reflections on integral human development can seem like fragments of spiritual direction for the Church faithful. Given the emphasis on social, ethical, and economic themes of integral human development, there is a need then to develop the "face" of the spiritual–theological dimension: to "grow in holiness." Accordingly, the article seeks to bring out a new direction of integral human development through a Levinasian lens, bringing the richness of the Jewish and Christian traditions together, namely, a spiritual–theological horizon for the imagination of faith to be articulated in service, responsibility, care, and love for others.

## 2. History and Truth

At the origins of developing a definition of integral human development is the concept of humanism and how it relates to history. Jacques Maritain reflects:

To leave the discussion quite open, let us say that humanism . . . tends to make man more truly human and to manifest his original grandeur by enabling him to participate in everything which can enrich him in nature and history . . . it demands that man develop his powers, his creative energies, and the life of reason, and at the same time labor to make the forces of the physical world instruments of his freedom" (Maritain 1936, p. 1).

The ideal of human nature is connected to transcendence, going beyond oneself to employ "powers" of "creative energies" and "reason." Freedom and truth signify in the "human" a "uniqueness as someone for whom no one else can substitute himself" (Levinas [1998] 1999, p. 59). Suppose freedom itself is to become "truly" a part of human goodness through the generosity of alterity, a life of holiness demonstrated by the Beatitudes (Matt 5–12). In that case, one may wonder what contaminates the "creative energies" and "life of reason" to be "inflated and altered" (Levinas [1998] 1999, p. 59) by the ego. For Maritain, people are tempted by error. By revealing such a position, he invites reflection on the foundation of Christian humanism (and integral human development):

Instead of an open human nature and an open reason, and this is real nature and real reason, people make out that there exists a nature and a reason isolated by themselves and *shut up* in themselves, and exclusive of everything, not themselves.

Instead of a human and rational development in continuity with the Gospel, people demand such a development as replacing the Gospel (Maritain 1936, p. 2).

Integral human development is otherwise than the self-centered ego that possesses a "history" of replacing the Gospel with practices and ideas that separate them from God's love and, hence, salvation history. Walter Kasper points out the theological consequence of denying the "Kingdom of God" (the "age of love") (Kasper [1977] 1993, p. 86):

Love reveals itself as the meaning of life. The world and man find fulfillment only in love. In practice, however, human beings have separated themselves from the love of God by sin and put themselves at the service of egotism, self-seeking, self-will, self-advantage, and self-importance. Everything falls apart in meaningless isolation and a general battle of all against all. In place of unity come loneliness and isolation, and the isolated individual falls victim to meaninglessness. . . . Love is the answer to the search for a just and human world, the solution to the riddle of history. It is the wholeness of man and the world (Kasper [1977] 1993, pp. 86–87).

If "modern atheistic humanism asserts" that God is a "restriction . . . of human freedom" (Kasper [1977] 1993, p. 16), then the consequences are ominous: the search for a humanism of "liberation and reconciliation" purged of its Judeo–Christian "identity," which must therefore demand another history, not a "salvation history," but one of enlightenment (reason) denying religious faith, or, for example, the barbarism of political and nationalistic ideologies that speak more of torture, murder, and hatred. For Levinas, the essence of modern anti-humanism is that it makes the human person rather than God "the aim of reality" (Levinas 2003, p. 56). Reflecting on the "essence of evil" as the sin and egoism of refusing to be responsible (denying that God has immemorially or anarchically made/ordered/ordained a human person to be responsible), Levinas writes:

From a responsibility even more ancient than the *conatus* of substance, more ancient than the beginning and the principle, from the anarchic, the ego returned to self, responsible for others, hostage of everyone, that is, substituted for everyone by its very non-interchangeability, hostage of all the others who, precisely *others*, do not belong to the same genus as the ego because I am responsible for them

without concerning myself about their responsibility for me because I am, in the last analysis and from the start, even responsible for that, the ego, I; I am man holding up the universe "full of things." Responsibility or saying prior to Being and beings, not saying itself in ontological categories. Modern anti-humanism may be wrong in not finding for man, lost in history and in order, the trace of this pre-historic an-archic saying (Levinas 2003, p. 57).

The progress of appreciating the spiritual import of integral human development rests in divine love. In the age of love, in the Kingdom of an unthematizable God beyond ontological assertions made in egoism, violence, and hostility, there remains an ethical-spiritual hope to "introduce sense into being" (the Levinasian idea of having a sense in being) and consciousness (Levinas 2003, p. 56). Such anarchic or "pre-original responsibility" directs the heart and soul of the human person to a new "knowing" and "wondering," namely to understand the temptation to separate oneself "from the Good" (Levinas 2003, p. 55). St. Paul knew this spiritual reality where, in paraphrasing Isa 64:4, he reflected:

> But, as it is written, 'What no eye has seen, nor ear heard, nor the human heart conceived, what God has prepared for those who love him'—these things God has revealed to us through the Spirit, for the Spirit searches everything, even the depths of God. For what human being knows what is truly human except the human spirit that is within? So also, no one comprehends what is truly God's except the Spirit of God. Now, we have received not the spirit of the world, but the Spirit that is from God, so that we may understand the gifts bestowed on us by God. And we speak of these things in words not taught by human wisdom but taught by the Spirit, interpreting spiritual things to those who are spiritual. (1 Cor 2:9–13)

If indeed the Kingdom of God is the "age of love," the Levinasian notion of having a sense of being and taking hold of the inner vocation of the "uniqueness of the I" means that one cannot escape responsibility (Levinas 1996a, p. 55). To have a sense of the existence and reality of being is to encounter a "persecuted truth" (Levinas 1998b, p. 55) in history, that the I is hostage (responsible) to the other. The metaphor of a "hostage" means a grave responsibility to hold up the universe with God and, hence, to take on a hyperbolic responsibility of love (agape). Such a "persecuted truth" points to the ideals of holiness and humility: "an unexceptional responsibility" (Levinas [1998] 1999, p. 59) to be "humble, as allied with the vanquished, the poor, and the persecuted" (Levinas 1998b, p. 55).

One may imagine that truth is a function of timidity or weak passivity. Rather, truth is otherwise than the essence of prejudice, judgments, and infantile beliefs that reduce others to facts, cold objectivity devoid of ethical subjectivity, inflating the worth of historical data with tempting illusions of truth. Beyond such errors of historical constructs beholden to political, economic, and social representations and even violence, there is the analogy of being par excellence in Jesus Christ. His compassion and courage provide a pathway toward integral human development. Pope Francis reflects:

> Look at Jesus. His deep compassion reached out to others. It did not make him hesitant, timid, or self-conscious, as often happens with us. Quite the opposite. His compassion made him go out actively to preach and to send others on a mission of healing and liberation. Let us acknowledge our weakness, but allow Jesus to lay hold of it and send us tooon mission. We are weak, yet we hold a treasure that can enlarge us and make those who receive it better and happier. Boldness and apostolic courage are an essential part of mission (Pope Francis 2018, no. 131).

To "acknowledge our weakness," the meekness and gentleness to "inherit the earth" (Matt 5:5; cf. Ps. 37:11) signifies an awakening towards "the abundant prosperity" (Ps. 37:11) of being formed in mission by Christ. This is to receive a "treasure" of "boldness and apostolic courage." Integral human development lies in God's depths, in the Spirit that submits to God apophatically (1 Cor 2:9–10), in order to understand spiritual wisdom other

than the being of the essence and historical representations that remain resistant to the face of the poor and stranger. "Proximity to God" is a function of humility and even persecution (Levinas 1998b, p. 56). To approach God and grow in holiness is also to wrestle with the darkness of the world. Where history and truth collide together, leaving a dust storm, so to speak, in its wake of uncertainty and ambiguity, one may discover settled amongst the scattered particles, almost unseen, glimpses, visions, and voices of evil and death.

### 3. Truth beyond Evil and Death

The horrors of human life put faith and belief into question. Along the spiritual path of integral human development, healthy doubt will orient the soul along a journey of disturbance and awakening. To survive the encounter of death in its evil forms of war, murder, alienation, hatred, and horror, one must first see the other's face in all its humanity: weakness, rage, suffering, hurt, and pain. How does one survive the horror of a world bent on personal and social evil? How does one find the courage and boldness to build "a good society" (Pope Benedict XVI 2009, no. 4)? The danger of relativism can damage the human soul by surgically removing, as it were, from it the inner search for meaning, truth, and wisdom. It is not surprising that Maritain asserts, "And for human life, for the concrete movement of history, this means real and serious amputations" (Maritain 1936, p. 2). Forms of beauty, goodness, and truth transformed by charity are thus disformed by indifference in the soul, resulting in attempts to emaciate charity, misconstrue it, and empty it "of meaning" (Pope Benedict XVI 2009, no. 2). It then becomes then a mission of maturity and boldness to link charity and truth together, as St. Paul demonstrates:

> We must no longer be children, tossed to and fro and blown about by every wind of doctrine, by people's trickery, by their craftiness in deceitful scheming. But speaking the truth in love, we must grow up in every way into him who is the head, into Christ, from whom the whole body, joined and knitted together by every ligament with which it is equipped, as each part is working properly, promotes the body's growth in building itself up in love. (Eph 4:14–16)

Noting St. Paul's emphasis on "truth in love," Pope Benedict points to the "inverse and complementary sequence of *caritas in veritate* [love in truth]" (Pope Benedict XVI 2009, no. 2). Pope Benedict explains: "Truth needs to be sought, found, and expressed within the 'economy' of charity, but charity in its turn needs to be understood, confirmed, and practiced in the light of truth" (Pope Benedict XVI 2009, no. 2). Here, we find a good context for a spiritual appreciation of integral human development as a means to bring theory and practice together. There is a mutual interrelationship or indwelling between truth and charity/love. By beginning in truth, we understand that humility and even persecution are modes in which the world of the other and of God come to mind. The truth in love is discovering the call to holiness. By practicing love, the ideal of service in the community and world, the light of truth can shine through goodness and beauty, revealing unity and the glory of God as peace and healing in the world. Hence, love in truth signifies a pathway to growing in holiness.

The integral human development of mission and vocation reveal together a charism or gift of boldness (*parrhesía*), which Pope Francis states is a form of "holiness" (Pope Francis 2018, no. 129). Moreover, he reflects:

> Parrhesía is a seal of the Spirit; it testifies to the authenticity of our preaching. It is a joyful assurance that leads us to glory in the Gospel we proclaim. It is an unshakeable trust in the faithful Witness who gives us the certainty that nothing can "separate us from the love of God" (Rom 8:39).

Holiness in the face of relativism and the horror of evil is a vocation of witnessing to the practice of "love in truth" as much as the search for "truth in love." The evils of the world, the indifference to human suffering, war, and forms of hatred such as oppression, economic corruption, and social prejudice produce a murderous relativism that gives "up God" (Maritain 1936, p. 3). Levinas speaks of this as the doctrine of "primitive powers"

or "elementary force" of "secret nostalgia" (Levinas 2004, p. 13) of nationalism, that is to say, "Hitlerism" (Levinas 2004, p. 15) in all its diabolical forms yesterday and today. In an epigraph written at the beginning of Levinas' major work, *Otherwise than Being*, he writes, "To the memory of those who were closest among the six million assassinated by the National Socialists, and of the millions on millions of all confessions and all nations, victims of the same hatred of the other man, the same anti-Semitism" (Levinas [1998] 1999). The existence of evil, or "Hitlerism," has many forms. A spiritual theology of integral human development, if it is going to embrace its vocation of witnessing "love in truth" as "God is love" (1 Jn 4:16), must not be afraid to think about (and respond to) evil and death. Speaking with boldness, even to the extent of awakening the soul of Christianity in its mission of charity in truth, Levinas reflects on one of the darkest times of history:

> Hitlerism is more than contagion or folly; it is an awakening of elementary emotions.
>
> And that makes it terribly dangerous and philosophically interesting. Because elementary sentiments harbor a philosophy. They express the primary attitude of a soul faced with the whole of the real and its own destiny. They predetermine or prefigure the sense of the soul's adventure in the world.
>
> So the philosophy of Hitlerism overflows the philosophy of Hitlerians. It casts doubt on the very principles of a civilization. This is not limited to conflict between liberalism and Hitlerism. Christianity itself is endangered, notwithstanding considerations or concordats granted to the Christian Churches at the advent of the regime (Levinas 2004, p. 13).

There is no witness of love without wounds, without remembering the horror of the past, without seeking a new world, an age of justice, mercy, forgiveness, and reconciliation. If baptized Christians seek a political and nationalistic "adventure" while amputating the Gospel and face of Christ and the poor one, then the "eros" and "intoxication" (Pope Benedict XVI 2009, no. 4) of hatred for the other will soon consume the soul into a terrible abyss: the negative choice to be against God. The pull of "elementary emotions," such as hatred, fury, hostility, rudeness, resentment, or the frenzy of murderous intentions animated by anger, fear, and madness, signifies a short journey toward evil and death.

Otherness, as another mode of truth, relates to a choice other than taking on the impulses of such elementary emotions. Otherness plays out for the soul as an ethical drama, a trauma of encountering the suffering other who puts the conscience into question. The boldness of otherness disturbs the abrasive life lost in abhorrent rationalism. This is because otherness breaks open from the depth of [de profundis] our existence with a moment of agape. The sense here of "de profundis" directs the soul towards the Infinite God. Because love, in its patience, "does not insist on its own way" (1 Cor 13:5), it embraces the vocation of the Spirit to search "everything, even the depths of God" (1 Cor 2:10). Levinas has pointed out gravely that "Christianity is endangered" where it makes false covenants ("concordats") with evil. In almost psalmic prose (compare Psalm 137), the Jewish novelist Chaim Potok (1929–2002) laments:

> On 1 September 1939, the legions of Germany invaded Poland.
>
> Rivers ran with blood. Beasts roamed the land. Men slew children. People fled from towns for the security of the forest. Blood and fire and pillars of smoke concealed the sun and fused together into an enormous apocalypse, an auto-da-fé that burned the soul of the world.
>
> Who was King? Who was not King? Centuries of Christian and Roman hate were king in the guise of Teutonic pagans. The world was as silent as the Nile had once been. And the wanderings of European Jewry came to an end (Potok 1978, pp. 512–13).

Christian humanism had failed to take hold in Europe. An apocalypse ensued: war in all its horror and the death, torture, and starvation of millions. In 1936, Maritain observed,

"prayer, miracle, supra-rational truths, the idea of sin and grace, the evangelical beatitudes, the necessity of asceticism, of contemplation, of the means of the Cross—all this is either put in parenthesis or is once for all denied. In the concrete government of human life, reason is isolated from the supra-rational" (Maritain 1936, p. 3). Here, Maritain has pointed out some key ingredients, so to speak, for holiness and boldness in a world bent on the annihilation of others. The "supra-rational" of faith, truth, and charity, essential for Christian humanism, stands as a call to take a stance against any form of apocalypse besieging the world. Such a call takes on an eschatological horizon of hope or patient endurance (Rev 2:3): a journey into "the depths of God."

## 4. Into "The Depths of God"

A spiritual theology of integral human development initiates a vigilance to take on God's wisdom and the first fruit of the Spirit, namely love (Gal 5:22). The "manifestation of the Spirit," aimed at "the common good" (1 Cor 12:7), leads the person of faith towards "the depths of God" (1 Cor 2:10), a "weeping" (Ps. 37:1) of compassion in spirit. Here, we encounter the heart of integral human development and an opportunity to speak of spiritual theology through an ethical, metaphysical lens. For the life of the soul, "de profundis," "from the depths of God," has a double meaning: (i) vulnerability towards the other and (ii) the vulnerability of human fragility inside the soul (Burggraeve 2023). The vulnerability towards the other helps to reveal what we are called prophetically to do (love in truth), while the Infinite Other, God, enters uniquely into the depths of our lives, orienting our vulnerability, ensoulment, and weakness towards a redemptive road (truth in love). This is a road to "the depths of God," an acceptance of the Christ who can rise within our wounds and tears, pointing to new directions, new life, or even a retreat back to God as a means to jump further into grace.

Being vulnerable and responsible towards the other identifies a crack or wound, a weakness "to inherit" (Matt 5:5) from God and the other's word. Vulnerability that forms into integral human development orients the self from within and outside towards a practice of charity, service, and responsibility. As finite human beings, we fail in our relationships with others. We are sinful beings and ethically imperfect because we are slow to realize what we should do or be. Life is like a Shakespearean comedy tragedy. The comedy is that we suddenly realize we are responsible, yet the tragedy is that we are too late (Levinas 1998a, p. 166). In Levinas' language, illeity, the hidden trace of God, becomes confused with the anonymous and depersonalizing evil of the "stirring of *there is*" (Levinas 1998a, pp. 165–66). We stumble in our responsibility for the other as "the laughter sticks to one's throat when the neighbor approaches—that is, when his face, or his forsakenness, draws near" (Levinas 1998a, p. 166).

Ethical imperfection reveals the need to grow in the spontaneity of faith. In the limitation of the bodiliness of hope, there is the failing of the vigilance of love because the human ethical striving to "be for the other" requires healing, redemption, consolation, patience, and humility. Regarding integral human development, this means taking on the spiritual practice of opening oneself in prayer to grow in holiness. Such a prayerful disposition is one with an ethical life (sensibility, touchability, and vulnerability). Where ethics and prayer come together, there is the risen Christ imparting the Holy Spirit (Jn 20:19–23), a new perspective of grace coming to consciousness and being. In Levinasian terms, this illustrates having a diachronic sense of the immemorial depths of God in human existence. Here, as the depth of our soul encounters ensoulment by the other (who is the soul of our soul), a "dis-inter-estedness" [rupture among beings] ethically "undoes" a person's "esse" [self-interested "being"] (Levinas 1998c, p. 90). In other words, in the other's face, one hears "the Word of God" (Levinas 1998b, p. 110). In a Christian theological sense, one may suggest here that Christ is resurrected in us ("a believing seeing" (Kasper [1977] 1993, p. 139)) or that we become touchable through the Spirit to orient our journey to the Father's Kingdom. To encounter the risen Christ is to have our senses, thoughts, and emotions opened towards faith in the hope that ethical living and prayer become one.

The meaning of "dis-inter-estedness" can be placed into greater ethical proximity and clarity, where Levinas puts into service Martin Buber's and Franz Rosenzweig's exegesis of the Golden Rule (Levinas 1998c, p. 90). Levinas points out that Buber and Rosenzweig were "perplexed" about translating the Hebrew "*kamokhah*" (as yourself) as it relates to the "you" in the Golden Rule, "you shall love your neighbor as yourself" (Lev 19:18; Matt 22:39). They read it "otherwise" so that the "you" becomes more the form of bodiliness of the other (a compassionate unity with the other): "Love your neighbor; this work is like yourself"; "Love your neighbor; he is yourself"; "it is this love of the neighbor, which is yourself" (Levinas 1998c, p. 90). Levinas goes further, employing the biblical method of a Dominican priest by reading the verse here in terms of the context of the whole book of Leviticus. Hence, Levinas provides a probing search to understand how ethics precedes ontology. In the concern for the other, there is a dissymmetrical relation of vulnerability that evidences an opening towards what "no one has heard, no ear has perceived" (Isa 64:4; cf. 1 Cor 2:9), namely a multitude of faces of the other. Hence, Levinas writes:

> Now, in the entirety of the book, there is always a priority of the other in relation to me. This is the biblical contribution in its entirety. And this is how I respond to the question: "Love your neighbor; all that is yourself; this work is yourself; this love is yourself." *Kamokhah* does not refer to "your neighbor," but to all the words that precede it. The Bible is the priority of the other [*l'autre*] in relation to me. It is another [*autrui*] that I always see the widow and the orphan. The other [*autrui*] always come first. This is what I have called, in Greek language, the dissymmetry of the interpersonal relation. If there is not this dissymmetry, then no line of what I have written can hold. And this is vulnerability. Only a vulnerable I can love his neighbor" (Levinas 1998c, p. 90).

From what Levinas suggests here, there is no vulnerability without encountering others' wounds, tears, and suffering. This is also the bodiliness and compassion of the self's wounds, tears, and suffering. For example, in bodiliness, one suffers through the suffering of the other or is wounded through the wounds of the other. This means that in the stumbling of responsibility, the vulnerability of gentleness (meekness) unveils a hidden, sacred wound (illeity) in which God may enter with a gift of agape (charity) to orient responsibility for the neighbor, the stranger ("alien"), widow and orphan (Lev 19:34; Deut 22:21–34), "the poor, the crippled, the blind, and the lame" (Lk 14:21). Entering into the vulnerability of love as agape, the self encounters the depths of God, "the secret of sociality," to respond like Moses, "Here I am" (Ex 3:4) (Levinas 1998b, p. 131). In such "extremes of gratuitousness and futility, love of my neighbor, love without concupiscence," God's word in the other's cry for mercy and help signifies that the "fear for the death of my neighbor is my fear" (Levinas 1998b, p. 131).

Perhaps one may have to walk decades to hear such voices with greater ethical and spiritual clarity. This is because there is a diachronic working of grace to make time ethically driven by the "de profundis" word of God coming to mind in the other's face. An incarnational encounter emerges of the word becoming flesh: the Holy Spirit ordering/ordaining the divine word to become flesh from the infinite depths of God, proclaiming Jesus the Christ as God and in God, as one of us, and in the profound depths of the soul. This suggests, in terms of the relation of Christian beliefs to the Golden Rule, that to love the neighbor "as yourself" is to encounter "charity in truth, to which Jesus Christ bore witness by his earthly life and especially by his death and resurrection" and how this is "the principal driving force behind the authentic development of every person and all humanity" (Pope Benedict XVI 2009, no. 1). Moreover, Pope Benedict qualifies this statement by adding, "Love—caritas—is an extraordinary force which leads people to opt for courageous and generous engagement in the field of justice and peace. It is a force that has its origin in God, Eternal Love, and Absolute Truth" (Pope Benedict XVI 2009, no. 1).

Love is an "extraordinary" force to crack the brittle world of sin. The truth of love reveals the animating force of hope for salvation and eternal life. Here, it is appropriate now to invite reflection on a theology of hope to develop a spirituality of "love in truth,"

of suffering for the other "because God's love has been poured into our hearts through the Holy Spirit that has been given to us" (Rom 5:5). Eschatology has a role to play in a spiritual theology of integral human development, one that can surprise the formation of consciousness with the depths of God, an abiding hope to unmask and conquer the evils of the world with moments of peace, unity, and love that have the "extraordinary force" to last through the decades of life and centuries of human history.

### 5. Towards a Theology of Hope

The nature of doing eschatology through explorations and debates can appear to be contentious and risky, especially where theologians attempt to offer new explanations and theories about "the end of the human person and of history" (Kelly 2006, p. ix). In the attempt to develop eschatological ideas about the end times and immortality, temptations may arise to surround the articles of faith with apologetics and rationalizations that seem to go beyond what "revelation permits" (von Balthasar 1998, p. 13). In other words, where the conception of eschatology entertains speculations around doctrine and dogma, there remains the danger of falling into an "abstract void" (von Balthasar 1998, p. 14) about the mystery of the end of an individual's life and the "consummation of the cosmos" (Kasper 1985, p. 377). Eschatology betrays an inherent risk of falling into "pseudo-logical speculations" (von Balthasar 1998, p. 14). Yet, eschatology is also the stuff of "patient endurance" (Rev 1:9; 14:12) and humility (a function of truth in love) before the mysteries of faith. Moreover, there exists a certain affectivity and vulnerability (integral human development) in eschatology: "an astonished stammering as we circle around the mystery on the basis of particular luminous words and suggestions of Holy Scripture" (von Balthasar 1998, p. 13). Here, the searching believer and theologian alike can take courage and confidence to journey towards hope seeking understanding, greater clarification, and intelligence of pressing questions about the "final reality" (eschaton), namely "the last things" (eschata), such as death, the intermediate state, heaven, hell, the Parousia and "the parable of hope" (the paschal mystery) (Kelly 2006, p. 21).

In terms of the mystery of eternal life, the 1992 International Theological Commission (ITC) document "Some Current Questions in Eschatology" presents a reflection on damnation, stating, "Death in the Lord implies the possibility of another way of dying, namely death outside the Lord, which leads to a second death (cf. Rev 20:14). For in this death, the power of sin through which death entered (cf. Rom 5:12) manifests to the fullest extent its capacity to separate us from God" (ITC 1992, no. 6.3). The ITC's eschatological statement on "death outside the Lord" offers an opportunity to think otherwise about integral human development and the potential of human sin, namely to consider what can be done about human evil and human acts that could lead to the annihilation of the person and soul, to the "second death" of hell. To grow in holiness is also to witness to the "lost" (Lk 15:24) and "spirits in prison" (1 Peter 3:19) to encourage hope for "repentance" (Lk 15:7) and new life.

Developing a humanism of integral human development, Maritain asserts:

Having given up God so as to be self-sufficient, now man is losing track of his soul, he looks in vain for himself, he turns the universe upside down, trying to find himself, he finds masks, and behind the masks death.

And there comes a spectacle which we witness: an *irrational* tidal wave. It is the awakening of a tragic opposition between life and intelligence (Maritain 1936, p. 3).

A spiritual theology of integral human development is challenged to seek an eschatological horizon of questioning. This does not mean, in any way, departing from the world of the living. The threat of hell and the danger of the "second death" command a challenge to take the consequences of evil seriously and, hence, to find a way so that love and justice might bring liberation, salvation, and truth to humanity tempted and haunted by the "evil," "horror," "anxiety," and "fear" "in being" (Levinas 1995, pp. 19–20). To begin, for integral human development to take on an eschatological character, is to listen to the words of the Risen Christ to "Go therefore and make disciples of all nations, baptizing them in the name

of the Father and of the Son and the Holy Spirit, and teaching them to obey everything that I have commanded you" with the assurance that "And remember, I am with you always, to the end of the age" (Matt 28:20). The mission of Christ through the Church unveils liberation and salvation: "Authentic, complete liberty is impossible without the primary liberation from death and perishability (sarx), from the power of sin, and from the law" (note also the "elements of the world"). 'It is with this freedom that Christ has set us free' [Gal 5:1]" (ITC 1976, no. 3). Authentic integral human development grows in boldness to know the "powers" (intelligence, absurdity, "ambiguity," senselessness, and seduction) of sin and evil (Levinas 2003, p. 56):

> Liberation from these powers, however, brings a fresh freedom, in consequence of which we can, in the spirit of Jesus Christ, be effective in love so as to serve our brothers and sisters. Here surely we have a foreshadowing of what God will himself accomplish as his gift to the just when he judges the whole story of humankind. The justice of God, through the Spirit and by his power, bestows a liberating action that enables us to work what is good, an action that finds its perfection through love (ITC 1976, no. 3).

A "sense" and sensibility of hope for salvation in "being" (Levinas 2003, p. 56) in a world falling into the threat and horror of evil and destruction can be communicated in theological terms as a paschal, a Trinitarian encounter of revelation: the death and resurrection of Christ evoke the presence of the Spirit, signifying the Father's love and justice "for us" (1 Jn 4:16). Such an encounter further signifies the force of Parousia, the eternal gift of salvation. The work of such a transformative encounter and awakening evokes a divine interruption of sin and evil. The light of the Parousia communicates "holiness," "boldness" (*parrhesía*), and "seal of the Spirit," the divine energy, as it were, to testify to the Gospel (Pope Francis 2018, no. 129). The glory and light of the Parousia work their way into human history and time as the divine interruption contradicts the brittle shell of sin:

> Man's shell is not hard enough, however, for it is formed of a contradiction. Perhaps the man whose shell can be broken is not yet really in hell but only—in his rebellious attitude to God—turned towards it. 'Therefore it is said,' says Paul: 'When he ascended on high he led captivity captive' (Eph 4:8, AV); the Lord 'takes it back with him and by this means releases men from their captivity to sin', thereby 'fashioning this very alienation from God into a way of approach to him' (von Balthasar 1998, pp. 312–13).

In the contradiction of God finding a way through human alienation and godforsaken, the mystery and surprise of divine interruption unfold, a second chance, so to speak, for the evildoer to hear Christ's first words, "as disruptive as new wine in old wineskins" (Mk 3:22): "What [or Whom] are you looking for?" (Jn 1:38; 20:15). In the process of divine interruption, the "what" of the sinner's decision (to reject God) transforms paradoxically into the "whom" of the personal encounter with the "ultimate strange attractor" (Kelly 2006, p. 186), the "Lamb of God who takes away the sin of the world" (Jn 1:29). Christ, the "strange attractor," embodies the covenantal heart of Levinasian ethical transcendence and truth: "The one for the other" (Levinas [1998] 1999, p. 165). Christ reveals that he is the divine miracle and light of God in humans. Christ is the source and outpouring of redemption, responsibility, justice, love, and compassion—a "treasure" before fragile vessels (2 Cor 4:7). Moreover, Christ is a "strange" Other, in the absolute uniqueness and character of mission, death, and resurrection, is the interrupting and surprising force of Parousia (love in truth). This suggests as Roger Burggraeve points out in his reflection on the parable of the Good Samaritan through the philosophy of Emmanuel Levinas, "'The absolutely foreign alone can instruct us'; (TI [*Totality and Infinity*] 45/73). We do not find the other in ourselves, the other comes to us as a revelation, that means as a master and a calling" (Burggraeve 2020, p. 279). Christ, the "strange attractor" and "master" of mercy, offers the new wine salvation, demonstrating the absolute passivity and contradiction of the light and boldness of Parousia.

Through such divine "calling" and interruption, there remains hope to renew the image of God within the sinner. This identifies an eschatological sense of mission to orient integral human development to begin new steps to grow in holiness to be a person in Christ.

The inescapable presence of Christ, who is "more absolutely' forsaken" than others in his death and descent to hell on Holy Saturday, calls into question humanity's "apparent, pretended inaccessibility" (von Balthasar 1998, pp. 312–13). For the forsaken one lost in hatred and the evil of sin, appearing godforsaken, meeting the Christ in a moment of grace, of love in truth, reflects the character of the Parousia as the perfection of boldness affirming, '"Do not be afraid" (Mk 6:50) and "I am with you always, to the end of the world" (Mt 28:20). In such vigilance of "surplus of responsibility" (as expiation and substitution), Christ witnesses to a "proximity" of the trace of illeity from the depths of God's immemorial love (Levinas [1998] 1999, p. 100). Christ, whose humility, poverty, and compassion are beyond any need for reciprocity and in whom "the whole fullness of the deity dwells bodily" (Col 2:9), stretches the communion of the hope that not "one of these little ones should be lost" (Matt 18:14), for "there will be more joy in heaven over one sinner who repents than over ninety-nine righteous people who need no repentance" (Lk 15:7).

If the forsaken ones begin to awake out of an anonymous existence of evil, to be interrupted and surprised by Christ before them (truth in love), then hope remains to contradict human sin. In such paschal integral human development, the suffering of Christ becomes the catalyst to have knowledge of God and of oneself (von Balthasar 1989, p. 261). This then leads to an encounter with the subjectivity of Christ's being (divine personhood) becoming one with the objectivity of Christ's being of love. By introducing a "sense" of love in their "being" (human existence) (Levinas 2003, p. 56), the unity of Christ's objective identity as savior and his subjective experience of suffering for others opens the theological imagination of faith. It leads to the contemplation of a paschal, spiritual element of integral human development: that the face of Christ interrupts the being of sinners by contradicting and disarming the horror of their "anonymous" and "impersonal" existence (Levinas 1995, p. 82) with the gift of love in truth (mercy) and truth in love (vulnerability). This revelation exposes the "unpredictable" reality that "human existence is never a realm of total control" (Kelly 2006, p. 186).

No one can successfully hide from God's love and mercy. Sin is but a brittle shield. In the moments of life and death, there remains the hope and possibility of transformation for all to die "in the Lord" (ITC 1992, no. 6.3) through encountering Jesus' unconditional and surprising promise of forgiveness. For in Christ, the "strange attractor," there lies hope to testify to the light of salvation: the Parousia and Kingdom of God reveal that "God is love" (1 Jn 4:16). In such integral human development, the reality of divine love appears like an overwhelming surprise, breaking the shell of sin and pricking the inflated balloon of the self on the way to damnation.

## 6. Conclusions: To Be Sons and Daughters of God

Jacques Maritain uses the metaphor "the intelligence of the serpent" to reflect on "counter-humanism," namely the dark forces against integral human development (Maritain 1936, p. 4). Frustrated and disturbed, he refers to the "terrible voices" of a "mediocre and base multitude" taking the form of "apocalyptic signs" by way of employing "the hatred of reason" (Maritain 1936, pp. 4–5). The "terrifying voice" of the Serpent's intelligence, of evil and lies, takes form in "the cult of the fecundity of war or in that cult of race and blood" (Maritain 1936, p. 5). In a prophet's voice, Maritain laments, "We've had enough of lying optimism and illusory moralities, enough of hypocritical justice and hypocritical right, enough of liberty that starves workmen and burns the stack of grain, enough of idealism that does us to death, which denies evil and unhappiness and robs us of the means of struggling against them . . ." (Maritain 1936, p. 4).

Writing in the midst of Hitlerism (1933–1945) and its worship of the "way of slaughter" and "lower powers," Maritain was beginning to witness and foresee the impending catastrophe awaiting humanity, the death of Jews in "concentration camps" and "Europe

maddened in an armament race and feverishly preparing for suicide," and the rise of Marxism, in which "reason . . . decapitates reason" (Maritain 1936, pp. 5–6). Such was the "horror of the night 'with no exits'" in the twentieth century, where humanity reduced divine transcendence to the false transcendence of "being in general," namely "anonymous being," an "impersonal form" of depersonalizing existence that "leads us to the absence of God" (Levinas 1995, pp. 45, 57–58, 63). Levinas names such horror as the "there is" [*il y a*] (Levinas 1995, p. 60). He most likely would characterize Maritain's reflection that the "evil in being," of "counter-humanism," has become "the evil of being" and hence that which transforms people into "evildoers," "impersonal" specters of existence, "disturbing themselves like phantoms" as they exist without any sign or trace of ethical subjectivity (Levinas 1995, pp. 19, 61). Humanity has witnessed that the dark forces opposing integral human development have varied human faces, such as the totalities of war, nationalism, individualism, and rejection of God.

In response to the evil that continues to plague the world, finding new guises with technology and the advancement of intelligent humanity—the serpent's intelligence, so to speak—there is the disarming witness of the New Testament to be sons and daughters of God. "This means," according to Maritain, the evocation of a "spiritual childhood" (Maritain 1936, p. 7):

> . . . in the spiritual order, the discovery of the *ways of spiritual childhood* whereby the "humanity of God our Saviour," as Saint Paul says [Titus 3:4], finds, with fewer human trappings, a readier way into man, and causes more souls to enter into this dark hidden task of suffering and vivifying; it implies in the moral and social order, the discovery of a deeper and fuller sense of the dignity of the human person, so that man would re-find himself in God re-found, and would direct social work toward an heroic ideal of brotherly love, itself conceived not as a spontaneous return of emotions to some illusionary primitive condition, but as a difficult and painful conquest of the spirit, of the work of grace and virtue (Maritain 1936, pp. 7–8).

A spiritual theology of integral human development suggests that "the humanism of the Incarnation" (Maritain 1936, p. 8) sets no limits for God to enter humanity. The vulnerability of God breaks open the divine "de profundis," giving the life of the soul a moment of grace and love, an invitation to be born of God (Jn 1:13) in the divine light of "goodness and loving-kindness" (1 Jn 1:4). To be the children, sons, and daughters of God is to take steps to learn to approach, appreciate, understand, and live out the mystery of the Incarnation: "But to all who received him, who believed in his name, he gave power to become children of God . . ." (Jn 1:12). To live by and through the Incarnation signifies how, in the depths of God, one is called to be personal, alive, and forgiving: "See what love the Father has given us, that we should be called children of God; and that is what we are" (1 Jn 3:1). Moreover, Zimmerman points out, "Christian praxis, framed as it is within the context of the theology of Incarnation, will struggle to credibly articulate a philosophical basis for its own ethical action without a corresponding account of the other, in which the giftedness of the body is given its due import" (Zimmermann 2009, pp. 992–93). Might this point to an integral humane ecology of the soul to be "lived out joyfully and authentically" (Pope Francis 2015, no. 10), a life of the bodiliness of the "redemptive incarnation" (Pope John-Paul II 2003, no. 58)? Or even a process of discovery, from "adoration of God" to "a deeper wonder" at oneself being led by Christ, patiently and sensitively, "to the full truth" (Pope John Paul II 1993, no. 8)?

The spiritual and theological formation of integral human development is not a process of self-directed education fuelled by "those forms of ersatz spirituality—having nothing to do with God—that dominate the current religious marketplace." (Pope Francis 2018, no. 111). Rather, integral human development invites an "incarnated passivity" or bodiliness of openness towards God, "an impossibility of evading" a life of "expiation" and "substitution" for the other (Levinas [1998] 1999, p. 112) that one encounters, for example, in the promise of forgiveness of the other. To conquer and "overcome the evil one," to know the

Father's forgiveness, suggests a spiritual praxis of taking a stance towards a world tempted by "the desire of the flesh, the desire of the eyes, the pride in riches" (1 Jn 2:12–15). This is not simply to make a person "better" despite "the Good it expresses," since it "perhaps makes all our discussion suspect of being 'ideology'" (Levinas [1998] 1999, p. 93). To embrace rather the newness of God's gift of charity and forgiveness suggests possessing the confidence of "boldness before God" (1 Jn 3:21) to be a person in Christ. One then comes to realize that at the heart of integral human development, there is the accusation/inner reflection (that which makes us responsible to the point of substitution for others (Levinas [1998] 1999, p. 112)), culminating in the question, "How does God's love abide in anyone who has the world's goods and sees a brother or sister in need and yet refuses help?" (1 Jn 3:17).

The focus of our eternal life in Christ should not be removed from our present encounter with history (Maritain 1936, p. 8). An eschatological existence of hope (just as much as love and faith) demonstrates a reference point for what remains "integral" to human development. The concern of the other is "integral" to faith in God as it becomes a test to witness that "God is love" (1 Jn 4:16), serving as much as the incarnational truth, "the word became flesh and lived among us, and we have seen his glory, the glory as of a father's only son, full of grace and truth" (Jn 1:14).

In a practical way, such an "incarnational" life is exemplified by the metaphor of being "in one's skin" and hence responsible without evasion and "recourse to anything" (Levinas 1996a, p. 89). People, lost in self-interest, can fall into "indolence" and, further, "fatigue" (Levinas 1995, pp. 29–31), paralleling the dispositions of acedia and spiritual torpor. A spiritual theology of integral human development has the boldness and courage to awaken the children of God (truth in love) towards a passivity of divine goodness and desire that aims for peace in this world (love in truth). Levinas explains in terms of deconstructing evil, such as war:

> Peace therefore cannot be identified with the end of combats that cease for want of combatants, by the defeat of some and the victory of the others, that is, with cemeteries or future universal empires. Peace must be my peace, in a relation that starts from an I and goes to the other, in desire and goodness, where the I both maintains itself and exists without egoism (Levinas 1996b, p. 306).

Levinas' ethical, metaphysical approach to understanding peace as a function of otherness (illiety of God), a "beatitude" of goodness and holiness, signifies the formation and transcendence of the "I" beyond "egoism." Such "integral human development" is "a driving force of charity in truth" (Pope Benedict XVI 2009, no. 77), unveiling the Spirit's fruit of "peace" (Gal 5:22): "Peace I leave with you; my peace I give to you. I do not give to you as the world gives. Do not let your hearts be troubled, and do not let them be afraid" (Jn 14:27). The peace that Christ gives is the truth that gives freedom (Jn 8:32) beyond the impulses, nationalisms, and hunger for death that turn the desire for the good and charity into projects of horror and world suicide.

Accordingly, a spiritual theology of integral human development possesses an eschatological vocation and mission to proclaim, "Peace, peace, to the far and the near, says the Lord; and I will heal them" (Isa 57:19). In other words, the divine calling for the sons and daughters of God is to live a responsible life to build up "the *earthly city* in unity and peace" (Pope Benedict XVI 2009, no. 7). In a spirit of waiting for Parousia, this reflects "to some degree an anticipation and a prefiguration of the undivided city of God" (Pope Benedict XVI 2009, no. 7). Such "anticipation" is oriented by the divine generosity of God's Spirit, the "largesse-in-us" (the emotional sensibility overflowing to the soul from the depths of God) of loving one's neighbor.

Following Descartes, Levinas writes with exceptional insight on the "intellectual feeling" of generosity ("disinterestedness"): "Descartes speaks with generosity. He attaches it both to the 'free disposition of [a man's] will' and to the fact that those who are generous 'do not hold anything more important than to do good to other men and to disdain their individual interests'" (Levinas [1998] 1999, pp. 157–58). What is remarkable here is that

Levinas adopts a Cartesian position rather than a Kantian one that fails to acknowledge "goodness . . . charity and mercy" (Levinas [1998] 1999, p. 158) in sensibility [*Sinnlichkeit*].

Levinas' position offers the insight that ethics and prayer come together where sensibility animates the moral conscience to listen to "the will of God" (Levinas [1998] 1999, p. 158) to grow in holiness. Hence, integral human development beholden to God's will (the incarnational divine logic of the Word becoming flesh in Jesus the Christ (Jn 1:14)) abides in the sensibility of love and truth: growing in holiness. This suggests that integral human development unveils a life otherwise of formation, otherness, and vulnerability, aiming towards discovering peace and unity in the world (truth in love) through, in, and for Christ. In a final word, integral human development testifies to a praxis of prayer, generosity, and holiness (love in truth), as Pope Francis attests: "We are called to be contemplatives even in the midst of action and to grow in holiness by responsibly and generously carrying out our proper mission" (Pope Francis 2018, no. 26).

**Funding:** This research received no external funding.

**Institutional Review Board Statement:** Not applicable.

**Informed Consent Statement:** Not applicable.

**Data Availability Statement:** Not applicable.

**Conflicts of Interest:** The author declares no conflict of interest.

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
