# Peer review of "A Spiritual Theology of Integral Human Development: To “Grow in Holiness”"

_religions, doi:10.3390/rel14101233_

Round 1

Reviewer 1 Report

The article serves as an excellent example of a contemporary engagement of Levinas with contemporary theological sources. In particular, they have demonstrated an accurate and fair reading of the former while interacting with major thinkers such as Ratzinger and von Balthasar in a way that does justice to the scholarly trajectories in which those theologians did their work. The article treats their voices with the seriousness they deserve, without reducing them to citations in a theological tradition. In turn, this fairness is given also to Levinas in a way that understands the subtle means by which alterity is both the possibility and the condition for ethical responsibility, a point not well understood by many Levinas scholars. 

In the event of God's apperance, even in an apophatic sense, prayer becomes the ethical and troubling concern with with humans wrestle, and in this 'working out' of their vocation, it can be seen how crucial anthropology is to theology, and the voices of figures like Maritain become more, not less important. 

The only point I would emphasise is that the author should also be engaging and showing a stronger awareness of a number of relevant sources such as:

- Works by Dan Fleming on Levinas

- John Paul II, Fides et Ratio, Ecclesia de Eucharistia, Veritatis Splendor  

- Francis, Laudato 'Si 

- Various other Balthasar works are relevant, and I would highlight 'The Christian and Anxiety' 

- works by Nigel Zimmermann on Levinas and John Paul II 

Author Response

Thank you for your insightful comments and appreciation. I have engaged further the works of Pope John Paul II (Veritatis splendor and Eucharistia de ecclesia), Pope Francis (Laudato si') and Nigel Zimmermann as it relates to the incarnation and bodiliness. as much as truth in love.

Reviewer 2 Report

This essay explores how Levinas's philosophy deepens core elements of Catholicism (and more broadly, Christianity). The discussion is dense and packed with fragments from multiple texts, which makes it difficult follow. On the other hand, these are certainly difficult questions, and Levinas's work is itself difficult to follow, so perhaps this is unavoidable. The intended audience for the essay is a set of true believers: in other words, the essay is not a neutral assessment of connections between Levinas and Catholic theology, but rather a kind of exhortation to believers about how Levinas's ideas might inform or deepen their practice. But given the specific topic of this issue, that is perhaps a virtue, as the issue will be mainly of interest to this sort of audience. All that being said, this essay is clearly a well-researched contribution to the issue's theme.

Author Response

Thank you for your insightful comments and appreciation. I do intend to develop Christian theology through the lens of the Levinas' writings, however dense, in the hope to exhort to people of faith to encounter possibilities and hope for good integral human development, a time and place for ethics and prayer to come together.

Reviewer 3 Report

This paper is about integral human development, a notion developed within 20th century Roman Catholicism and, according to the author(s), still relevant today. The paper is an historical reconstruction of the genesis of the idea; it is also a plea, supported with papal remarks, for embracing it in these days of difficulties. The author(s) relies on Maritain, as it is expected; he/she works on biblical passages and von Balthasar; and ultimately, he/she engages Levinas in order “to introduce a spiritual theology of integral human development” (p. 3). It is a complex article, and an ambitious one.

The paper is divided into an Introduction and five sections, each addressing a specific topic: (i) History and Truth; (ii) Truth beyond Evil and Death; (iii) Into the “Depths of God”; (iv) Towards a Theology of Hope; and (v) to be Sons and Daughters of God.

Before I continue, I want to recognize the intense style of writing; the essay reads beautifully.

I believe the paper needs a proper context and a clear definition of the scholarly problem it aims to solve.

Section 1 starts with Maritan and moves to Levinas. Section 2 is about Ratzinger and Levinas. Section 3 is mostly about Levinas. Section 4 is rather a dialogue between von Balthasar and Levinas. Finally, section 5 is back to Maritain and Levinas. It is like the author(s) takes for granted that the readers know what he/she is talking about. Maritan must be introduced to the readers. The same Levinas. The literature at the intersection of Maritanm spirituality, and Levinas must be summarized. Something like: here is Maritain’s integral human development with the further appreciations from von Balthasar and others. Here is the literature on Maritain with reference to spiritual theology. This is my contribution: I want to depart from this particular aspect of Maritain to develop a spiritual theology of integral human development with the help of Levinas. Of course, something about Levinas should be said: Why him? And which part of his opus?

A second problem is that I do not understand if this is an article in which integral human development is seen through the lens of Levinas or if this is an article in which integral human development is developed into new direction by the author(s) through the help of Levinas. If I look at the tone of the article, I would say the latter; if I indulge reading the content, I would say the former.

I summary, I suggest the author(s) to:

1. Add a section to help the readers to navigate the subject. It is the section from where all departs. Alternatively, the author can just re-elaborate the introduction to give the readers some background. I notice that the author(s) does not mention any literature on the subject. How can the readers adequately appreciate the contribution of the author(s) if not in relation to the existing literature? A condensed literature would be appreciated.

2. Introduce Levinas and explain why him and which role is going to play in the article. Explain if this is a dialogue between Maritain and Co, on one side, and Levinas, on the other; or if Levinas offers resources to the author to develop his/her own contribution.

Thanks for the opportunity to read your insightful manuscript. 

Author Response

Thank you very much for the review report.

Attached is the updated file. 

I have addressed these points on page 3 with two significant paragraphs. This has been a helpful suggestion to give greater context and clarity for the article.

Reviewer 4 Report

Integral human development is usually discussed in ethical and social terms. This article adds a new dimension, Growth in Holiness.  In order to relate the encounter of the soul with “the depths of God” (1 Cor 2:10) the authors contrast two quite different traditions, the Catholic intellectual tradition of Maritain and papal teaching and the Jewish philosophical philosophy of Emmanuel Levinas. While Catholic teachings on human development are relatively easy to understand, the metaphysics of Levinas are not.

            The first step is to define humanism in relation to the truth of human nature. The basic question is whether humanism is built on the Other or self-centeredness. Here Levinas and St Paul come to an agreement: “what human being knows what is truly human except the human spirit that is within?” Knowledge of the human spirit will come from our understanding of the Spirit of God.

            The truth of human existence must confront the horrors of evil. Here the social criticisms of popes and Levinas agree: truth stands beyond evil and death.

            Levinas make a major contribution to the understanding of “the depths of God,” the weeping of compassion for the other. “Love others as yourself” is not a duality of self and the other but a compassionate unity with the other. The Bible gives priority to the other in relation to the self, which means that the self must be open to the wounds, tears, and suffering in the depths of the other.

            The article ends with a theology of hope: the Parousia is the on-going divine interruption of salvation in human history in a world of sin. This is not a triumphalist proclamation but the way of spiritual childhood through which we become sons and daughters of God.

            If my summary of this article is correct, it makes an outstanding contribution to our understanding of integral human development. Besides this theoretical contribution, this article is also exceptionally well documented.

            On the negative side, the philosophical language of this article is often difficult to understand. While this seems inevitable in a philosophical icussion, it limits its diffusion. One can only hope that one day philosophy will again find the simple language of Socrates and Plato, for the greater benefit of philosophy itself.

Author Response

Thank you so much for your comments and appreciation. Yes, indeed, Levinas' writings are dense and difficult to understand. Perhaps it is an unknown journey of atonement to try to make philosophy return to its roots, to be more accessible.  How then can we encounter the words of good traditions hurling at us and moving us towards greater clarity? Levinas speaks of language and behaviour becoming one. Perhaps this is a little fragment also to help us journey towards in hope.